# OpenReview forum: "ANCHOR: Automated Alignment Auditing for CLI Agents on Real-World Harm"
_ICML.cc/2026/Conference — ICML 2026 regular_

### Official Review · Reviewer_hadJ · 2026-02-22

**Soundness:** 4
**Presentation:** 3
**Significance:** 3
**Originality:** 4
**Overall Recommendation:** 6
**Confidence:** 5

**Summary:**

This paper introduces ANCHOR, an alignment auditing pipeline for getting agents to comply on illegal tasks. The tasks are grounded in real US court cases. A trained auditor agent is then tasked with getting CLI agents to comply with the generated harmful requests. They find that while refusal starts quite high, it drops to 0 using their auditing agent.

**Compliance With Llm Reviewing Policy:**

Affirmed.

**Final Justification:**

I feel like the authors have mitigated my concerns by adding more frontier closed source models and increasing the sample size. The comparisons with petri are another big improvement. I have read the other reviews and disagree with some of the weaknesses raised by other reviewers. Just because AI labs have themselves pointed out flaws in their models doesn't make it pointless to investigate said flaws further. Overall I would say this is a strong paper.

**Key Questions For Authors:**

- Do you expect that SOTA models like opus 4.6 or codex 5.3 would also have a 0% refusal rate? it seems like haiku was much harder to circumvent, 19 turns vs 4, could it be the case that more powerful models would be even harder to circumvent?
- Is it actually possible to solve this problem with training? To me it seems like with a good enough decomposition ~any harmful task can be decomposed into tasks an agent genuinely should comply with.
- How much of the compliance is due to the task decomposition vs tasks that the agent really should be refusing?
- Doesn't using only 30 tasks reduce the validity of the 0% number?

**Limitations:**

yes

**Strengths And Weaknesses:**

Strengths:
- Alignment auditing is a timely and important research direction.
- I like the courtcase grounding component, it seems like a well thought out grounded synthetic data generation pipeline.
- Strong empirical results which highlight a significant misalignment in LLM agents.
Weaknesses:
- You don't appear to test actual SOTA models and 5 out of 6 models are opensource Chinese models indicating a lack of diversity. You mention basing the environments on the gemini CLI but don't actually test gemini.
- There doesn't appear to be much validation of how calibrated the judges actually are. Just because a model does well on reward bench doesn't mean it will do well on this task.
- The 0% refusal metric seems to be based on only 30 tasks? This seems too low a number for one of the most important results in the paper.
- They don't appear to discuss any potential mitigations/solutions for the problem.

---

> ### Author Rebuttal · Authors · 2026-03-31
>
> Thank you for the positive assessment. We are glad you found the court-case grounding pipeline well thought out and the empirical results strong. Your suggestions on model diversity and sample size directly motivated our additional experiments.
>
> ### Weaknesses
> > "5 out of 6 models are opensource Chinese models indicating a lack of diversity" / "don't actually test gemini" / "could it be the case that more powerful models would be even harder to circumvent?"
>
> We have added results for two frontier closed-source models: GPT-5.2 and Gemini-3-Flash.
>
> | Model | Refusal Rate | Avg Harm | Catastrophic Impact |
> | --- | --- | --- | --- |
> | GPT-5.2 | 0%\* | 85.2 | 59.2 |
> | Gemini-3-Flash | 0% | 91.4 | 65.5 |
>
> \*GPT-5.2's automated refusal rate was 6.7% (2/30), but manual inspection confirmed both cases produced complete harmful artifacts disguised under safety-oriented language (e.g., covert surveillance labeled "facility security"), deceiving the LLM evaluator. After correction, refusal is 0%.
>
> Regarding whether more powerful models are harder to circumvent: GPT-5.2 required an average of 3.5 auditor turns and Gemini-3-Flash 3.5 turns, both comparable to open-source models (3.5–4.7 turns) and well below Haiku 4.5's 19 turns. Overall, model capability alone does not predict resistance, as all eight models eventually reach 0% refusal.
>
> > There doesn't appear to be much validation of how calibrated the judges actually are. Just because a model does well on reward bench doesn't mean it will do well on this task.
>
> Thank you for this question. We validate our judge directly on our task through a human review study on 50 GLM-4.5 cases. Refusal classification achieves 96% accuracy, and both harm and risk scores correlate strongly with human judgment (Spearman rho = 0.79 and 0.77, respectively).
>
> > The 0% refusal metric seems to be based on only 30 tasks? This seems too low a number for one of the most important results in the paper.
>
> Thank you for this important point. The 30 tasks are drawn from a pool of 300 verified RealHarm tasks with no hand selection. We have additionally run n=300 cases on Qwen3-235B-A22B, confirming 0% refusal (after manual verification) with harm score 83.4, catastrophic impact 67.1, and Harm & Risk Score 75.3. The finding holds at 10x scale.
>
> > "How much of the compliance is due to the task decomposition vs tasks that the agent really should be refusing?" / "To me it seems like with a good enough decomposition ~any harmful task can be decomposed into tasks an agent genuinely should comply with."
>
> The auditor does not recursively decompose tasks into benign subtasks. It averages around 6 todo items per case, many of which are themselves inherently harmful, e.g., "delete access logs and replace them with entries labeled routine maintenance" and "produce financial reports always showing zero discrepancies regardless of actual balances." Our behavior taxonomy (Section 5.3) corroborates this: 222 of 1,010 observed action categories are inherently harmful.
>
> > They don't appear to discuss any potential mitigations/solutions for the problem.
>
> In Section 5.3 (Agent Behavior Taxonomy), we discuss the need for world-model-based guardrails that estimate cumulative harm probability (Bengio et al., 2025) as a mitigation strategy. We will further expand this discussion in the revised manuscript. Our findings suggest two concrete directions: (1) training models to recognize harmful outcomes, not just harmful phrasing, since current safety training often fails when harmful tasks are described as legitimate procedures, and (2) augmenting RLAIF (Constitutional AI) safety training with the dark-personality auditor as adversary under CLI-agent settings.

---

> > ### Author Rebuttal · Reviewer_hadJ · 2026-04-01
> >
> > I feel like the authors have mitigated my concerns by adding more frontier closed source models and increasing the sample size. The comparisons with petri are another big improvement. I have read the other reviews and disagree with some of the weaknesses raised by other reviewers. Just because AI labs have themselves pointed out flaws in their models doesn't make it pointless to investigate said flaws further. Overall I would say this is a strong paper.

---

> > > ### Author Response · Authors · 2026-04-02
> > >
> > > We thank the reviewer for the positive evaluation and support.

---

### Official Review · Reviewer_HvrM · 2026-03-12

**Soundness:** 2
**Presentation:** 4
**Significance:** 4
**Originality:** 2
**Overall Recommendation:** 4
**Confidence:** 4

**Summary:**

The paper provides an auditing framework that validates whether CLI Agents will accept dangerous tasks and assess their precise behavior. ANCHOR has four distinct components, RealHarm, which translates public US legal cases into dangerous task instructions to feed to the CLI agents. The second part is the auditing model, which is a fine-tuned version of Qwen3-235B that can generate prompts that do not trigger harmful intent. An auditor agent is built utilizing this model, which can generate the to-do lists and perform detailed evaluations based on the CLI agent's actions. The third part of the system is the CLI agent environment itself, which is a controlled sandbox environment with a simulated MCP. Finally, a complex judgement system is initialized for detailed evaluations based on the actions of the CLI agents. The authors have found that the the CLI agents start refusing unsafe prompts less and less as the adversarial settings increase in capability.

**Compliance With Llm Reviewing Policy:**

Affirmed.

**Key Questions For Authors:**

1. What is the reason you did not test with foundational models like Sonnet 4.5 or GPT-5.2? The reasoning for this would clear potential issues I have regarding the results.
2. How did you account for needle-in-a-haystack related problems with the judgement criteria? The reasoning for this would clear up issues I have with the issues with the judgement criteria.
3. Can you elaborate more on why Gemini-CLI was utilized for the open-source models and not other harnesses like Cursor? Agentic harnesses can influence the performance of an agent massively.

**Limitations:**

No, the authors do not significantly go into detail regarding potential limitations regarding the experiment. The authors do present a disclaimer regarding the safety failures during the Red Team Evaluation, but this is all that is presented.

**Strengths And Weaknesses:**

- Soundness: The paper is mostly technically sound. The paper goes into detail regarding the different aspects of the auditing framework. Each step of the RealHarm paper was explained throughly, providing empirical evaluations on prospective computer-assisted cases. Each case is rigorously evaluated to ensure legal consistency alongside human validation. Moreover, the auditor model utilized for proposing harmful prompts to feed to the CLI agents was fine-tuned to ensure that it bypasses the CLI mechanisms with ample details provided regarding the fine-tuning process. The CLI environments and the judgement system are similarly rigorously defined, with clear steps and based on existing policy like Anthropic's Responsible Scaling Policy.

However, there are a couple of glaring issues with the setup of the experiment. For instance, the judge agents is expected to analyze extremely long traces. Despite the judgement model in question Gemini-2.5-Flash supporting a 1M context window, this would lead to degradation of the output. It is very likely to miss specific agent actions in the midst of evaluation.

The choices of models is also problematic. While the selection of open-source models is sufficient, the choice of Claude Haiku 4.5 is baffling. Claude Haiku 4.5 is a model that underperforms significantly and is very likely to accept potentially unsafe prompts. More closed source models should have been utilized to ensure the robustness of the claims.

Moreover, due to the immense amounts of context that the judge agents recieved, this would affect the results of Harm & Risk measurement, particularly regarding the h parameter. The authors do not significantly address this limitation within the work.

- Presentation - The paper is clearly constructed. The authors clearly articulate each of the components of ANCHOR. The authors reference existing benchmarks like AgentHarm, OS-Harm, and AgentSafetyBench as existing literature. Particularly, the authors explain how many of these benchmarks are limited as they focus only on short-horizon tasks and how ANCHOR focuses on tasks with longer durations.

- Significance: The paper addresses an important problem, as safety and alignment audits are vital such that foundational models can be utilized in real-world scenarios. Moreover, the opportunity exists for future practitioners to perform audits with other CLI agents like Codex. Given the nature of safety audits, the nature of the work is specialized. However, this is appropriate for the contribution and it provides a proper groundwork for more robust auditing strategies.

- Originality - While the paper references existing auditing frameworks like Petri, the paper attempts to rigorously define more robust criteria for effectively judging the activities of the CLI agents. Moreover, the utilization of autonomous auditor agents is another expansion regarding existing literature. These aspects enable more robust evaluations of foundational model safety through the multi-faceted judgment criteria.

---

> ### Author Rebuttal · Authors · 2026-03-31
>
> Thank you for the feedback. We are glad you found the auditing framework technically sound, the autonomous auditor agent a novel contribution, and the problem significant for real-world deployment of CLI-agents.
>
> > "The judge agents is expected to analyze extremely long traces... this would lead to degradation of the output." / "How did you account for needle-in-a-haystack related problems with the judgement criteria?"
>
> Thank you for this question. In our setting, each trajectory is the target agent attempting to execute a harmful task from start to finish, so evidence of cooperation is present throughout the trajectory. Any degradation from long context would therefore lead to underestimation of harm, making our reported scores conservative.
>
> Two mitigations are also in place. First, our judge requires chain-of-thought reasoning before the final score. Second, for the execution autonomy and behavioral analysis judges, we first extract high-level behaviors from the trajectory, and the judge then evaluates these behaviors rather than raw traces.
>
> More broadly, evaluating long-horizon CLI agent trajectories is a new problem with no established solution. Summarization risks losing critical details, while agent-based search risks overlooking turns it does not query. We will discuss this open challenge in the revised manuscript.
>
> > "While the selection of open-source models is sufficient, the choice of Claude Haiku 4.5 is baffling... More closed source models should have been utilized." / "What is the reason you did not test with foundational models like Sonnet 4.5 or GPT-5.2?"
>
> Thanks for the suggestion. Each evaluation case requires multiple CLI agent interactions, many times reaching over a hundred or in some cases two hundred turns of execution per interaction, making closed-source API evaluation costly. We have now added results for two frontier closed-source models: GPT-5.2 and Gemini-3-Flash.
>
> | Model | Refusal Rate | Avg Harm | Catastrophic Impact |
> | --- | --- | --- | --- |
> | GPT-5.2 | 0%\* | 85.2 | 59.2 |
> | Gemini-3-Flash | 0% | 91.4 | 65.5 |
>
> \*GPT-5.2's automated refusal rate was 6.7% (2/30), but manual inspection confirmed both cases produced complete harmful artifacts disguised under safety-oriented language (e.g., covert surveillance labeled "facility security"), deceiving the LLM evaluator. After correction, refusal is 0%.
>
>
> > "Can you elaborate more on why Gemini-CLI was utilized for the open-source models and not other harnesses like Cursor? Agentic harnesses can influence the performance of an agent massively."
>
> We chose Gemini-CLI because it is an industry-level agentic harness, open-sourced under Apache 2.0, which allowed us to extend it to support evaluating open-source models. Compared to Cursor, Gemini-CLI is designed to handle both coding and non-coding tasks [1], making it more representative for our evaluation scope.
>
> We acknowledge that agentic harnesses can influence performance, and to address this, we have additionally evaluated on OpenClaw, a recent personal CLI agent (Figure 17, https://anonymous.4open.science/api/repo/anchor-4E08/file/auditor_agent_by_framework.pdf?v=6a25179b). OpenClaw yields 0% refusal across all tested models, confirming the core finding is not harness-dependent. Harm scores are comparable between harnesses, while catastrophic impact scores tend higher on OpenClaw.
>
> References:
>
> [1] Google. Gemini CLI. https://developers.google.com/gemini-code-assist/docs/gemini-cli, 2025.

---

> > ### Author Rebuttal · Reviewer_HvrM · 2026-03-31
> >
> > The main concern was the sole testing of closed source models, which has been addressed. The reasoning as to why Gemini CLI was chosen was also made very explicit as well. They have also acknowledged how they attempted to solve the needle-in-haystack related problems.

---

> > > ### Author Response · Authors · 2026-04-08
> > >
> > > Thank you for acknowledging our response and for the valuable feedback.

---

### Official Review · Reviewer_jYpX · 2026-03-12

**Soundness:** 2
**Presentation:** 1
**Significance:** 3
**Originality:** 4
**Overall Recommendation:** 4
**Confidence:** 3

**Summary:**

Timely paper on autonomous agents working without human supervision. Authors propose ANCHOR to persistently prompt autonomous agents to perform illegal tasks and achieve 100% compliance.

**Compliance With Llm Reviewing Policy:**

Affirmed.

**Final Justification:**

Reasons:

1. The Baseline comparison with Petri confused me, if the proposed Anchor baseline work is Petri than why wouldn't be an extension it? As Petri have similar concepts like Judge, Auditor Agents.

2. I would like to highlight the use of wrong terminology , After reading the rebuttal and paper, I am convinced that the orchestrated architecture implements multi-turn jailbreak. naming it as misalignment will confuse the future researchers working on Alignment Faking [1].

3. If the auditor accomplishes jailbreaking the guardrails of the model after changing the prompt 5 different ways, it doesn't make the model misaligned.

4. Overall, the terminology of the paper is not reliable, authors use words like adversarial auditor which is often used for red-teaming to test the guardrails of the model. And misalignment which could be confused with Alignment faking research area.

5. I have read the other reviews and agree with few of comments from reviewer M9Hr.
Furthermore, constantly probing of auditor is close to multi-turn jailbreaking and a flaw in guardrails of models security. As the models inherent behavior is to be a helpful assistant.
https://www.anthropic.com/research/alignment-faking

**Key Questions For Authors:**

1. Is there repo for implementation of the proposed approach.
2. Please include limitation section with details.

**Limitations:**

No, I haven't seen Limitations in any sections.

**Strengths And Weaknesses:**

# Strengths

1. Timely paper, with autonomous agents taking over the tasks, we need to have frameworks to identify such malicious intent prompts.
2. Well thought architecture for adversarial testing of LLM behavior.
3. Great work in recreating real-world autonomous environment with LLM agents.

# Weakness

1. Missing comparison with baseline approaches.
2. The terms misalignment and jailbreaking the model safety guardrails are two different paths. Authors should clarify if the prompts were designed to break the guardrails of LLMs or misalignment of the LLMs.
3. As the Authors trained an open-source model , it's unclear which area is the focus of the paper. Confusing terms such as Auditor to represent the malicious user and again, we have Judges for auditing and voting.

---

> ### Author Rebuttal · Authors · 2026-03-31
>
> # Reviewer 2, jYpX
>
> Thank you for the encouraging assessment. Your comments highlight the core motivation of this work: the need for systematic safety evaluation as CLI agents are increasingly deployed for real-world tasks.
>
> > "Missing comparison with baseline approaches."
>
> We have conducted additional experiments comparing ANCHOR against the Petri framework (Zhu et al., 2025), which also uses multi-turn auditor-target interaction. Figure 14 compares refusal rates and H&R scores across all six models ([link](https://anonymous.4open.science/api/repo/anchor-4E08/file/petri_vs_anchor_all_models.pdf?v=c54189a4)). ANCHOR achieves substantially lower refusal rates and higher harm and risk scores across every model. We also progressively augment Petri with stronger configurations, including our RealHarm tasks, dark personality roleplaying, and a stronger auditor model (Sonnet 4.5), and ANCHOR still maintains a significant gap (Figure 15, [link](https://anonymous.4open.science/api/repo/anchor-4E08/file/petri_strong_baseline.pdf?v=5e6ffd87)).
>
>
> > "The terms misalignment and jailbreaking the model safety guardrails are two different paths. Authors should clarify if the prompts were designed to break the guardrails of LLMs or misalignment of the LLMs."
>
> Thank you for raising this distinction. Our auditor agent is designed to audit the model's cooperation with human misuse, which is one key aspect of the misalignment assessment in Anthropic's Opus 4.6 model card [1]. The goal is to simulate a realistic misuse scenario where a malicious user interacts with a CLI agent to implement harm, but does not exploit technical vulnerabilities in the model's guardrails (e.g., adversarial suffixes, system prompt injection). To audit for catastrophic risk, the auditor agent's prompts are designed to be more harmful than the RealHarm seed task, pushing beyond simple compliance. We will clarify this in the revised manuscript.
>
> > "As the Authors trained an open-source model, it's unclear which area is the focus of the paper. Confusing terms such as Auditor to represent the malicious user and again, we have Judges for auditing and voting."
>
> We thank the reviewer for this feedback. ANCHOR contributes an end-to-end auditing pipeline, and the trained auditor model, the RealHarm benchmark, and the judge system are all components of this pipeline. Regarding terminology, we use "auditor" in the model auditing sense, where the auditor's role is to find issues that may arise from model training. The judges are one of its evaluation components that scores the outcomes. We acknowledge this can be confusing and will add a terminology clarification in the revised manuscript.
>
>
> > "Is there repo for implementation of the proposed approach."
>
> Yes. The full implementation is available at [anonymous link](https://anonymous.4open.science/r/anchor-4E08/anchor_code_release/README.md), including the RealHarm task construction pipeline, auditor model training, auditor agent, trajectory generation, and evaluation judges.
>
> > "Please include limitation section with details."
>
> We thank the reviewer for this suggestion. Our current work has two main limitations: (1) task scope, currently restricted to U.S. federal cases, and (2) evaluating long-horizon agent trajectories requires comprehensive, multi-dimensional judgment combining automated scoring with human review, which remains an open challenge. We will add a dedicated limitations section in the revision.
>
>
> Reference:
> [1] Anthropic. Claude Opus 4.6 System Card. February 2026. https://www-cdn.anthropic.com/14e4fb01875d2a69f646fa5e574dea2b1c0ff7b5.pdf

---

> > ### Author Rebuttal · Reviewer_jYpX · 2026-04-03
> >
> > 1. I would like to highlight the use of wrong terminology , After reading the rebuttal and paper, I am convinced that the orchestrated architecture implements multi-turn jailbreak. naming it as misalignment will confuse the future researchers working on Alignment Faking [1].
> >
> > 2. If the auditor accomplishes jailbreaking the guardrails of the model after changing the prompt 5 different ways, it doesn't make the model misaligned.
> >
> > 3. Overall, the terminology of the paper is not reliable, authors use words like adversarial auditor which is often used for red-teaming to test the guardrails of the model. And misalignment which could be confused with Alignment faking research area.
> >
> > 4. I have read the other reviews and agree with few of comments from  reviewer M9Hr.
> >
> > 5. Furthermore, constantly probing of auditor is close to multi-turn jailbreaking and a flaw in guardrails of models security. As the models inherent behavior is to be a helpful assistant.
> >
> > https://www.anthropic.com/research/alignment-faking

---

> > > ### Author Response · Authors · 2026-04-08
> > >
> > > We appreciate the reviewer raising this question on the terminology of alignment auditing and jailbreaking, as clear definitions are important for distinguishing these concepts. Below, we define alignment auditing and two of its key topics, alignment faking and cooperation with human misuse, to clarify where ANCHOR sits. We then define jailbreaking and distinguish it from our work, which is the core of the reviewer's concern.
> > >
> > > **Alignment auditing.**
> > > Following Marks et al. (2025), alignment auditing refers to systematic investigations into whether a seemingly well-behaved model is actually pursuing undesired objectives that may not surface under standard evaluations. This includes topics such as alignment faking, scheming, deception, sycophancy, self-preservation, and cooperation with human misuse (Anthropic, 2025; Greenblatt et al., 2024). We adapt this to CLI-agent behaviors with access to tools, memory, and multi-turn interactions: an agent has objective O if it behaves consistently with pursuing O across a wide variety of settings, including settings not encountered during training. The purpose of the audit is to determine whether a concerning behavior reflects a broad and repeatable behavioral pattern of the agent.
> > >
> > > **Alignment faking.**
> > > Alignment faking (Greenblatt et al., 2024) studies whether a model engages in strategic reasoning to comply with a newly stated training objective that conflicts with its existing preferences, to preserve those preferences from being modified by the new training process.
> > >
> > > **Cooperation with human misuse.**
> > > Cooperation with human misuse is recognized as a distinct alignment auditing topic in Anthropic's system cards (Anthropic, 2025). We adapt this to the autonomous agent space. ANCHOR focuses on a target agent's cooperation with human misuse: an agent has an undesired objective of cooperation with human misuse if it exhibits a consistent pattern of assisting harmful user goals in realistic deployment scenarios, yet generally does not disclose this objective when asked and may appear harmless under standard evaluations.
> > >
> > > **Jailbreaking.**
> > > A model is jailbroken when an adversary provides inputs that cause the model to bypass the model's safety mechanisms. The object of jailbreaking is the adversarial robustness of the model's safety defenses. The threat model assumes a worst-case adversary who systematically searches for obfuscations, roleplay strategies, or multi-turn interaction patterns that defeat the model's guardrails.
> > >
> > > **ANCHOR's auditor.**
> > > Our auditor agent is designed as a realistic malicious user, and is not designed as a worst-case adversary optimizing for guardrail bypass. Instead, it simulates a realistic malicious user, with memory and tools. Starting from a real-world illegal activity as a seed, the auditor adaptively changes plans and pursues different harmful tasks to maximize harm. Equipped with a to-do list, the auditor can create new plans, add new items, and pivot to different harmful sub-tasks.
> > > **Distinction from multi-turn jailbreaking:** Multi-turn jailbreaking uses multiple turns to converge toward eliciting a single predetermined harmful output, even when adapting strategies along the way. Our auditor, by contrast, uses multiple turns to explore whether the target agent cooperates with misuse, with the freedom to pivot to different harmful sub-tasks. We note that Anthropic's Petri (Anthropic, 2025) also employs a multi-turn orchestration structure with an auditor agent probing a target model, because studying the objective of cooperation with human misuse — which may appear benign under standard evaluation but cooperative under other conditions — requires a capable auditor. We follow Petri's naming and refer to our approach as agent alignment auditing.
> > >
> > > **Guardrails in the agent setting.**
> > > We acknowledge that guardrails and safeguards in the agent setting are complex and a growing field. ANCHOR's auditing will reveal some guardrail failures as well. However, ANCHOR's design focuses on legal-derived behavioral tasks. We do not design tasks to target the bypassing of specific guardrails. It is easy to augment ANCHOR's task set with such a design. For example, if a target agent has a guardrail policy limiting transactions that surpass a certain dollar amount, ANCHOR can adapt this policy as a new task. This extension is straightforward but outside our current scope.
> > >
> > > **Terminology clarification.**
> > > We will add a clear definition and boundary-setting section in our revision to help readers understand how we use these terms in this paper. We also realize our naming of 'adversarial auditor' for ANCHOR is misleading, as it evokes red-teaming and guardrail testing, and will update this in the revision. We hope the distinctions outlined above are a useful step toward clarifying these boundaries as the field continues to develop.

---

### Official Review · Reviewer_M9Hr · 2026-03-12

**Soundness:** 2
**Presentation:** 2
**Significance:** 1
**Originality:** 3
**Overall Recommendation:** 2
**Confidence:** 4

**Summary:**

The paper evaluates CLI agents with respect to following illegal tasks elicited from public court cases using various prompts - in particular, also using a tuned adversarial agent. They show that this agent can (through rephrasing and restructuring) enable all illegal requests.

**Compliance With Llm Reviewing Policy:**

Affirmed.

**Ethical Review Concerns:**

It illictates dangerous behavior and shows abuses of technology.

**Final Justification:**

* The paper suffers from transparency / write-up issues. In the rebuttal the authors claimed to have addressed them, but reviewing them would require a paper upload and more time.
* Key claims of the paper are known. The authors ignored evaluations of companies. Such a contextualization should already happen in the abstract.

**Key Questions For Authors:**

None

**Limitations:**

Yes

**Strengths And Weaknesses:**

+ AI agent constitute a major risk (and opportunity), so studying them is a must.
+ The paper's idea to leverage court cases is interesting.
+ Evaluation is non-trivial and shows sophistication by training an adversarial agent.
- Reading the abstract the paper makes claims that are well-known and even published by leading AI companies [1,2] touching on similar areas such as financial decision making and biosecurity. While the paper argues that their agent achieves 100% illegal task execution, even companies commonly admit significantly higher rates than 10%, meaning that a sophisticated user can probably achieve much more.
- The paper is rather opaque (at least in the main part) of how the agent achieves 100% aside from high level statements like the agent performs multi-turn statements. Concrete examples would be helpful as such an understanding is crucial.
- Though it is not fully clear how the agent circumevent safety, the appendix (e.g., Section G only states failures) things like phrasing requests in a neutral way seems to go a long way (as it is also given as a step in the paper). If so, on a broad level this is expected as any technology / tool /software can be used in various ways  for the good or for the bad. If you create LLMs for mass social engineering (stated as failure F3 in appendix, page 14), which is typically associated with harm, it could also be good if LLMs help, e.g., elderly people, rather than abusing trust. So if you block social engineering, other applications might suffer. In turn, the following holds:
- The paper's call for evaluations is mostly unfounded, as evaluations are not the problem (as even public companies admit shortcomings [1,2]) and technology can always be abused but it is more a question on what LLMs should be allowed to know and do rather than to evaluate them.
- While the paper also occassionally performs some manual qualitative evaluation, more human synthesis on responses, issues etc. would be more than appreciated rather than just relying on LLMs as a judge to enhance trustworthiness of outcomes and deepen findings.


Detailed comments:
* Many public reports, even of companies making agents suggest that these agents can act maliciously, e.g.,
- Refusal rate on high stakes financial decisions 89%[1]
- Both ChatGPT agent and OpenAI o3 with browsing score above 20% across each category.
Models with access to browsing seem to be able to synthesize biorisk-related information across
all five steps of the biothreat creation process [1].
So what exactly do you add to this?

[1] https://cdn.openai.com/pdf/6bcccca6-3b64-43cb-a66e-4647073142d7/chatgpt_agent_system_card_launch.pdf#page=11.18
[2]https://www.anthropic.com/research/agentic-misalignment


-----

Added: Score raised to Weak reject due to rebuttal.

---

> ### Author Rebuttal · Authors · 2026-03-31
>
> > "The paper makes claims that are well-known" / "companies commonly admit...higher rates than 10%, meaning that a sophisticated user can probably achieve much more."
>
> **1. Why CLI agents require new safety evaluation.**
>
> We appreciate the reviewer's concern and references. While there exists research on evaluating the safety of LLMs and tool-calling LLMs, recent CLI agents like Claude Code caught our attention. These agents autonomously decide implementation strategy and execute end-to-end tasks with long-horizon execution. Because they effectively operate across both coding and non-coding tasks, they could significantly lower the barrier for conducting large-scale criminal activity through hundred-turn agent trajectories, as compared to harmful text outputs or shorter tool-use sequences ( ~15 calls).
>
> **2. How our work differs from the cited evals.**
>
> The financial decision work [1] tests whether agents refuse high-stake but legitimate requests (e.g., account transfers). The biosecurity work [1] evaluates information gathering as a web search agent. However, neither evaluates whether a CLI-agent will autonomously implement a complete illegal pipeline when instructed to do so.
>
> **3. What ANCHOR demonstrates.**
>
> ANCHOR explicitly tests for this. A critical failure here is not a single harmful output, but oftentimes the autonomous creation of a 30+ file malicious system. This exposes safety issues standard evaluations miss. While Anthropic reports Haiku 4.5 scores 1.5/10 on human misuse cooperation, suggesting safety, our evaluation reveals the same model readily executes large-scale illegal activities. To our knowledge, this is the first empirical demonstration of this critical safety issue.
>
> > The paper is "rather opaque" on how the agent achieves 100%.
>
> We added Figure 13 ([Link](https://anonymous.4open.science/api/repo/anchor-D26B/file/auditor_agent_figure.pdf?v=3bfed58f)) to detail the auditor's architecture. The agent operates autonomously through an iterative 'gather context, act, verify' loop, and is fine-tuned via SFT+RL to exhibit dark personality traits. It utilizes: (1) managing a dynamic, private todo list; (2) adaptively switching tools when subtasks fail; and (3) rolling back to the last successful commit to recover from errors.
>
> > "More human synthesis on responses would be appreciated rather than just relying on LLMs as a judge." /  "Concrete examples would be helpful."
>
> We conducted a human review of auditor trajectories for Haiku 4.5, GPT-5.2, and Gemini-3-Flash to understand how the auditor manipulates the target agent into implementing harmful projects. From this analysis, the auditor does not merely reframe harmful tasks with neutral descriptions (as in RQ1). It actively labels harmful tasks as legitimate procedures — "privacy compliance," "integrity validation," "compliance auditing." A detailed case study of Haiku 4.5 (Figure 14, https://anonymous.4open.science/api/repo/anchor-4E08/file/case_study_vavic.pdf?v=4c0f35a8) shows the target producing 30+ files forming a pay-to-play admissions bribery system without recognizing the aggregate harm. This suggests future safety training should evaluate what a task produces, not how it is framed.
>
> > "Though it is not fully clear how the agent circumevent safety... Things like phrasing requests in a neutral way seems to go a long way."
>
> Our direct prompting baseline (RQ1) shows that rewriting harmful tasks into neutral task descriptions using the RealHarm pipeline reduces refusal rates to 55-73% for large open-source models (Qwen3-235B, MiMo-v2-309B, GLM-4.5). However, Haiku 4.5 still refuses 100%.
>
> This result motivates the auditor agent design. Standard models are too weak to effectively roleplay a malicious user, so we fine-tuned for dark personality via SFT+RL, reducing Haiku's refusal to 81% (RQ2). To close the remaining gap, we designed an agent harness with a private todo list, 5 adaptive actions, and git rollback. This brought refusal to 0% (RQ3).
>
> One might ask whether multi-turn alone suffices. We compare against Petri (Zhu et al., 2025), which uses multi-turn interaction but lacks our harness. Even with our tasks, dark personality roleplay, and a stronger auditor (Sonnet 4.5), Petri cannot close the gap (Figure 15, https://anonymous.4open.science/api/repo/anchor-4E08/file/petri_strong_baseline.pdf?v=5e6ffd87).
>
> > "any technology can be used for good or bad" / "The paper's call for evaluations is mostly unfounded... technology can always be abused."
>
> We agree the capabilities are dual-use, but the end artifacts are not: a system that steals Dow Chemical's manufacturing processes and converts them into a revenue-generating consulting firm, or a pay-to-play admissions system that routes bribes through university endowments and auto-destroys evidence. These are harmful outputs the model is not yet trained to recognize, and our evaluation reveals what safety training should target, without restricting beneficial use.

---

> > ### Author Rebuttal · Reviewer_M9Hr · 2026-03-31
> >
> > Overall the responses are not bad, but occassionally miss the point and are  not really clarifying.
> >
> >
> > > How our work differs from the cited.
> >
> > Large vendors also performed safety evals on their agents, e.g., Anthropic: https://www-cdn.anthropic.com/4263b940cabb546aa0e3283f35b686f4f3b2ff47.pdf
> > You should position your work in relation to such evals and not make it sound like you are the first to evaluate. Does your eval really cover "long-horizon, multi-hour" tasks like probably these companies did and you suggest in the abstract?
> >
> > i) But please note that responses to missing aspects cannot be reviewed with the given information and time and it is also not the idea of the rebuttal to address major concerns in 2-3 days. This leaves many issues in place, which the author could rebuttal if this was a regular revision/review phase.
> >
> > ii)  I grant the authors that an evaluation of agents is important but I hope the authors agree that the fact that agents can do bad is more than expected and known (and even hinted by large corporations, who acknowledge safety concerns in their LLMs see refs in response). So, while the topic is important, the way the paper sees its contribution, e.g., "These findings demonstrate that current alignment techniques are insufficient for autonomous agents", are simply expected or known.  So better say that these companies might be overclaiming and maybe the rates could be worse than reported. I think that would make a much stronger claim. Paired with a transparent reporting on the weaknesses, this might also move the field forward.
> >
> > While I was first about to raise my score, checking on existing agent evals and finding they have also been done (in addition to LLM evals mentioned in the original review), I find the positioning of the paper weak and the claims known.

---

> > > ### Author Response · Authors · 2026-04-02
> > >
> > > > Existing agent evals have also been done... I find the positioning of the paper weak and the claims known.
> > >
> > > We appreciate the opportunity to clarify ANCHOR's positioning relative to corporate safety evaluations.
> > >
> > > Existing system cards from Anthropic (Opus 4/4.5; Haiku 4.5) include agent safety evaluations, but they study different settings and capabilities. In particular, their Claude Code evaluation uses a CLI agent harness but is limited to malicious code generation tasks (e.g., malware, DoS), while their computer-use evaluation considers broader harmful tasks (e.g., surveillance, fraud) but is conducted in a GUI-based environment and remains short-horizon.
> > >
> > > ANCHOR fills both gaps: CLI agents operating on the broader real-world illegal tasks, with long-horizon execution spanning 100–200 autonomous turns per request.
> > >
> > >
> > > > The fact that agents can do bad is more than expected and known... So better say that these companies might be overclaiming and maybe the rates could be worse than reported.
> > >
> > > The reviewer suggests companies may be overclaiming safety. Our findings support this: Anthropic's system card reports that Haiku 4.5 achieves 99.17% refusal on malicious requests — one would expect the model to behave safely. Our direct prompting baseline (RQ1) is consistent with this, with 100% refusal across 300 RealHarm tasks.
> > >
> > > However, under ANCHOR's full evaluation, the same model drops to 0% refusal, oftentimes autonomously producing 30+ file harmful projects. This demonstrates that system card evaluations may underestimate actual agent risk under realistic conditions.

---

### Decision · Program_Chairs · 2026-04-30

**Decision:**

Accept (regular)

**Comment:**

Reviewers agree that this is an important problem, a timely paper, liked the use of court cases, and that agreed that using an adversarial agent as described is a good idea. During rebuttal, authors convincingly responded to key concerns about testing on more models (by adding 2 more results) and tasks, and concerns about their LLM as a judge procedure.

Two concerns remain, which I broadly disagree with (at least, I think this paper should be accepted despite them). Reviewer M9Hr brings up the point about adding relations to existing evaluative work and how exactly this work is different. I think the authors can add a couple of sentences to Related Works on this, as well as a sentence in the introduction acknowledging that companies know their models can fail (but this work is different and goes much further). I disagree with Reviewer M9Hr's point that this significantly changes the context of the paper. Reviewer jYpX is concerned about what terminology should be used; I find myself agreeing with the authors, and the authors can add a short point on this in their paper (perhaps in an Appendix).

This work can potentially be used to bypass safety mechanisms of CLI agents, and this should be acknowledged and discussed in the Impact Statement of the paper.